# FLASHBACK: UNDERSTANDING AND MITIGATING FORGETTING IN FEDERATED LEARNING

## ABSTRACT

In the realm of Federated Learning (FL), the convergence and effectiveness of learning algorithms can be severely hampered by the phenomenon of forgetting—where knowledge obtained in one round becomes diluted or lost in subsequent rounds. Such a challenge is a result of severe data heterogeneity across clients. Although FL algorithms like FedAvg have been pivotal, they often falter in scenarios of high data heterogeneity. This work delves into the nuances of this problem, establishing the critical role forgetting plays in the inefficient learning of FL in the context of severe data heterogeneity. Knowledge loss occurs in both the local update and the aggregation step; addressing one phase without considering the other will not mitigate forgetting. We introduce a novel metric that offers a granular measurement of forgetting at every round while ensuring that the occurrence of forgetting is distinctly recognized and not obscured by the simultaneous acquisition of new class-specific knowledge. Leveraging these insights, we propose Flashback, an FL algorithm that integrates a novel dynamic distillation approach. The knowledge of different models is estimated and the distillation loss is adapted accordingly. This adaptive distillation is applied both at the local and global update phases, ensuring models retain essential knowledge across rounds while also assimilating new knowledge. Our approach seeks to robustly mitigate the detrimental effects of forgetting, paving the way for more efficient and consistent FL algorithms, especially in environments of high data heterogeneity. By effectively mitigating forgetting, Flashback achieves faster convergence to target accuracy outperforming baselines, by being up to $88.5\times$ faster and at least $4.6\times$ faster across the different benchmarks.

## 1 INTRODUCTION

Federated Learning (FL) is a distributed learning paradigm that allows training over decentralized private data. These datasets belong to different clients that participate in training a global model. Federated Averaging (FedAvg) (McMahan et al., 2017) is a prominent training algorithm that uses a centralized server to orchestrate the process. At every round, the server samples a proportion of the available clients and then distributes to them the current version of the global model. Each client participant performs $E$ epochs of local training using their private data; and then sends back the updated model. Finally, the server aggregates the models by averaging them to obtain the new global model. This process is typically repeated for many communication rounds until a desired model performance is obtained.

A main challenge in FL is the heterogeneity in distribution between the private datasets, which are unbalanced and non-IID (Kairouz et al., 2019). Data heterogeneity causes local model updates to drift – the local optima are not consistent with the global optima – and can lead to slow convergence of the global model – where more rounds of communication and local computation are needed – or worse, when the desired performance may not be reached. Addressing data heterogeneity in FL has been the focus of several prior studies. FedProx (Li et al., 2020) proposes a proximal term to limit the distance between the global model and the local model updates, mitigating the drift in the local updates. MOON (Li et al., 2021b) mitigates the local drift using a contrastive loss to minimize the distance between the feature representation of the global model and the local model updates while maximizing the distance between the current model updates, and the previous model updates.

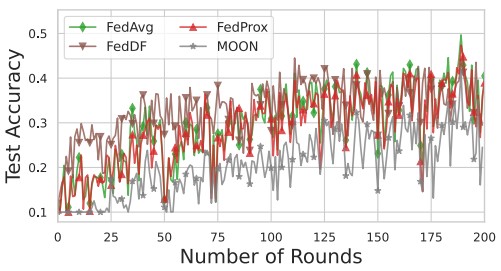 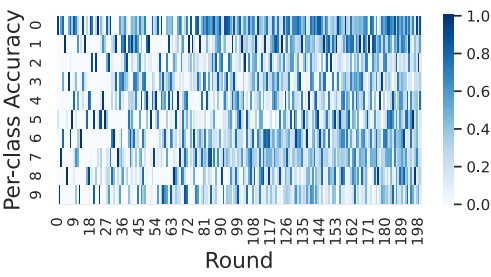

(a) Global model accuracy of FedAvg and baselines.
(b) Per-class accuracy of FedAvg's global model.

Figure 1: Performance of FedAvg and other baselines over training rounds with CIFAR10.

FedDF (Lin et al., 2020) addresses heterogeneity in local models by using ensemble distillation during the aggregation step at the server (instead of averaging the model updates).

However, we experimentally observe that under severe data heterogeneity, these proposals provide little or even no advantage over FedAvg. For instance, Fig. 1a illustrates the test accuracy of FedAvg and other baselines while training a DNN model with the CIFAR10 dataset (Krizhevsky, 2009) (details in § 5). Although this model could be trained at over 70% accuracy in a centralized setting, training with FL methods shows a slow and oscillating convergence, which barely reaches ≈50% accuracy after 200 rounds.

This motivates us to better understand how data heterogeneity poses a challenge for FL and devise a new approach of handling non-IID datasets. We investigate the evolution of the global model accuracy broken down by its per-class accuracy. Fig. 1b presents the per-class accuracy for FedAvg; each rectangle represents the accuracy of the global model on a class at a round. Other baseline methods show similar results. Our key observation is that there is a significant presence of *forgetting*: i.e., cases where some knowledge obtained by the global model at round $t$ is dropped at round $t+1$, causing the accuracy to drop (as shown by the prominent number of light-shaded rectangles at the right side of darker ones in the figure).

A similar phenomenon is known as *catastrophic forgetting* in Continual Learning (CL) literature (Parisi et al., 2019). CL addresses the challenge of sequentially training a model on a series of tasks, denoted as $\{T_1, T_2, \ldots, T_n\}$, without revisiting data from prior tasks. Formally, given a model with parameters $\theta$ and task-specific loss functions $L_t(\theta)$ for each task $T_t$, the objective in CL is to update $\theta$ such that performance on the current task is optimized without significantly degrading the model's performance on previously learned tasks. This is non-trivial, as naïve sequential training often leads to catastrophic forgetting, where knowledge from prior tasks is overridden when learning a new task. An inherent assumption in this paradigm is that once the model transitions from task $T_i$ to task $T_{i+1}$, data from $T_i$ becomes inaccessible, amplifying the importance of knowledge retention strategies (De Lange et al., 2021).

While the premises and assumptions of FL differ from those of traditional machine learning, forgetting remains an issue. This can be viewed as a side effect of data heterogeneity, a commonality it shares with CL. In FL, the global model evolves based on a fluctuating data distribution. Specifically, in each communication round, a diverse set of clients, each with distinct data distributions, contribute to the model update. This dynamic presents dual levels of data heterogeneity. Firstly, at the *intra-round level*, the heterogeneity arises from the participation of clients with varied data distributions within the same round. This diversity can inadvertently lead to "forgetting" knowledge from certain clients. Secondly, at the *inter-round level*, the participating clients generally change from one round to the next. Consequently, the global model might "forget" or dilute knowledge obtained from clients from previous rounds.

We propose Flashback, a FL algorithm that employs a dynamic distillation approach to mitigate the effects of both intra-round and inter-round data heterogeneity. Flashback's dynamic distillation ensures that the global model retains its knowledge during the local updates, and during the aggregation step by adaptively adjusting the distillation loss. Flashback performs these adaptations by estimating the knowledge in each model using label counts as a proxy. Overall, Flashback results in a more stable and faster convergence compared to existing methods.

Our contributions are the following:

• We investigate the forgetting problem in FL. We show that under severe data heterogeneity FL sufferers from forgetting. Then we show how and where forgetting happens in FL (§ 3).

• We propose a new metric for measuring global forgetting and local forgetting over the rounds in FL (§ 3).

• We introduce *Flashback*, a FL algorithm that employs a dynamic distillation during both local and global updates (§ 4). By addressing the forgetting issue, Flashback not only mitigates its detrimental effects but also converges to the desired accuracy faster than existing methods (§ 5).

## 2 BACKGROUND

We consider a standard cross-device FL setup in which there are $N$ clients. Each client $i$ has a unique dataset $D_i = \{(x_j, y_j)\}_{j=1}^{n_i}$ where $x_j$ represents the input features and $y_j$ is the ground-truth label for $j$-th data point and $n_i$ represent the size of the dataset. The goal is to train a single global model through the objective:

$$\arg\min_w \mathcal{L}(w) = \sum_{i=1}^N \frac{|D_i|}{|\cup_{i \in [N]} D_i|} L_i(w)$$

$L_i(w)$ represents the local loss for client $i$ and $l(w; (x_j, y_j))$ is the cross-entropy loss for a single data point, both defined as:

$$L_i(w) = \frac{1}{|D_i|} \sum_{j=1}^{|D_i|} l(w; (x_j, y_j)) \qquad l(w; (x, y)) = \mathcal{L}_{\text{CE}}(F_w(x), y)$$

where $F$ denotes the model function parameterized by weights $w$.

FedAvg provides a structured approach to this decentralized training. Starting with the global model $w_0$, it randomly selects $K$ clients from the available $N$ clients. In each round $t$, these chosen clients receive the previous global model, $w_{t-1}$, and optimize it based on their local data using $L_i(w_{k,t})$. Post-optimization, every client sends their updated model, $w_{k,t}$, back to the server. The global model is then updated by aggregating the model updates as:

$$w_t = \sum_{i=1}^K \frac{|D_i|}{|\cup_{i \in [K]} D_i|} w_{k,t}$$

To accommodate the intrinsic heterogeneity in client data, various FL algorithms introduce modifications either at the local update level or during the global aggregation. The nuances of these variations are further explored in § 6.

**Knowledge Distillation** is a training method wherein a smaller model, referred to as the student, is trained to reproduce the behavior of a more complex model or ensemble, called the teacher. Let $F_{w_s}$ denote the student model with weights $w_s$ and $F_{w_t}$ represent the teacher model with weights $w_t$. For a given input $x$, the student aims to minimize the following distillation loss:

$$\mathcal{L}_{\text{KD}}((x, y); w_s, w_t) = \alpha \mathcal{L}_{\text{CE}}(F_{w_s}(x), y) + (1 - \alpha) \mathcal{L}_{\text{KL}}(F_{w_t}(x), F_{w_s}(x)) \tag{1}$$

Here, $\mathcal{L}_{\text{CE}}$ is the standard cross-entropy loss with true label $y$, and $\mathcal{L}_{\text{KL}}$ represents the Kullback-Leibler (KL) divergence between the teacher's and the student's output probabilities. It is defined as $\mathcal{L}_{\text{KL}}(\boldsymbol{p}, \boldsymbol{q}) = \sum_{c=1}^C p^c \log\left(\frac{p^c}{q^c}\right)$, where $C$ is the number of classes, $\mathbf{p}$ is the target output probability vector, and $\mathbf{q}$ is the predicted output probability vector. The hyperparameter $\alpha$ balances the importance between the true labels and the teacher's outputs.

While distillation originally emerged as a method for model compression, its utility extends to FL. In the federated context, distillation can combat challenges like data heterogeneity Lin et al. (2020); Lee et al. (2021) and communication efficiency Jeong et al. (2018). Specifically, the global model can act as a guiding teacher during local updates, directing the training process for each client. Additionally, distillation techniques streamline the aggregation step, assisting in the incorporation of varied knowledge from diverse clients to update the global model. Furthermore, employing distillation for aggregation mitigates model heterogeneity, allowing for the use of different model architectures. Perhaps most interestingly, by using distillation, FL systems can potentially bypass the traditional method of transmitting weight updates. This is accomplished by sending soft labels that encapsulate the essence of local updates–communication becomes more efficient, reducing bandwidth usage.

## 3 FORGETTING IN FL

We now investigate where forgetting happens and devise a metric to quantify this phenomenon. Recall that in FL, the models are updated in two distinct phases: 1) during local training – when each client $k$ starts from global model $w_{t-1}$ and locally trains $w_{k,t}$ – and 2) during the aggregation step – when the server combines the client models to update the new global model $w_t$.

We observer that forgetting may occur in both phases. We refer to the former case as **local forgetting**, where some knowledge in the global model will be lost during the local training $w_{t-1} \rightarrow w_{k,t}$. This is due to optimizing for the clients' local objectives, which depend on their datasets. Local forgetting is akin to the form of forgetting seen in CL, where tasks (and clients in FL) change over time, and consequently, the data distribution. We refer to the latter case as **global forgetting**, where some knowledge contained in the clients' model updates will be lost during aggregation $\sum\{w_{k,t} \mid k \in S_t\} \rightarrow w_t$. This might be due to the coordinate-wise aggregation of weights as opposed to matched averaging in the parameter space of DNNs (Wang et al., 2020a).

We provide an illustration of forgetting in Fig. 2 based on actual experiments with several baseline methods. The figure shows the per-class accuracy of $w_{t-1}$, all local models $w_{k,t}$, and the new global model $w_t$. The local forgetting is evident in the drop in accuracy (lighter shade of blue) of the local models $w_{k,t}$ compared to the global model $w_{t-1}$. The global forgetting is evident in the drop in accuracy of the global model $w_t$ compared to the local models $w_{k,t}$. The figure also previews a result for our method, Flashback, which shows a significant mitigation of forgetting.

Moreover, local forgetting and global forgetting are intertwined, which means addressing the issue at only one of the phases will not be sufficient, since it will happen at the next phase, and therefore have a cascading effect into the same phase at the next round.

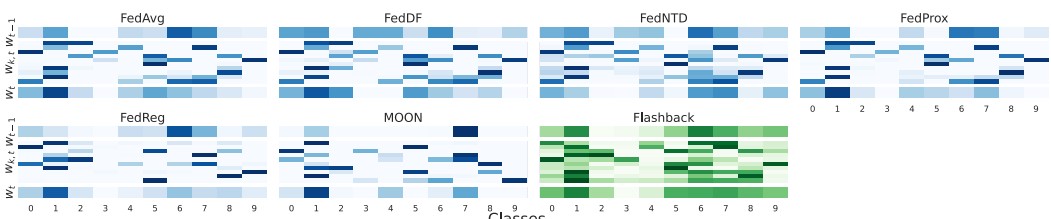

Figure 2: Local (client) & Global Forgetting in all the baselines using CIFAR10. The first row represents the global model per-class test accuracy at round $t-1$; then the rows in the middle are all the clients that participated in round $t$, and finally in the last row the global model at the end of round $t$. Local forgetting happens when clients at round $t$ lose the knowledge that the global model had at round $t-1$. The global forgetting happens when the global model at round $t$ loses the knowledge that in the clients' models at round $t$. Similarly results with other datasets are in Fig. 10.

In CL, forgetting is often quantified using Backward Transfer (BwT) (Chaudhry et al., 2018). Lee et al. (2021) adapted this metric for FL as:

$$\mathcal{F} = \frac{1}{C} \sum_{c=1}^{C} \arg\max_{t \in 1, T-1} (A_t^c - A_T^c), \tag{2}$$

where $C$ is the number of classes and $A_t^c$ is the global model accuracy on class $c$ at round $t$.

However, $\mathcal{F}$ is a coarse-grain score that evaluates forgetting in aggregate across all rounds. We seek a finer-grain metric that can measure forgetting at any given round. Further, we wish to account for knowledge replacement scenarios such as when a decline in accuracy for one class might be accompanied by an increase in another, essentially masking the negative impact of forgetting in aggregate measures. Thus, for our evaluation results (§ 5), we propose to measure forgetting across consecutive rounds by focusing only on drops in accuracy using the following metric:

$$\mathcal{F}_t = \frac{1}{C} \sum_{c=1}^{C} \min(0, -(A_t^c - A_{t-1}^c))$$

where $t > 1$ is the round at which forgetting is measured. Moreover, client $k$'s local forgetting $\mathcal{F}_{k,t}$ can be measured by substituting $A_t^c$ with $A_{k,t}^c$ (the accuracy of client $k$'s local model for class $c$).

## 4 FORGETTING ROBUST FL

Our key idea to mitigate both local and global forgetting is to leverage a dynamic form of knowledge distillation, which is fine-tuned in response to the evolving knowledge captured by the models. During local training, distillation ensures that each local model learns from the client's local dataset while retaining knowledge from the current global model. The aggregation step follows the approach of FedAvg to produce the new global model. However, this is followed by a distillation step where the new global model is treated as a student, learning from the previous version of the global model and the ensemble of local model updates, which are treated as teachers. Similar to the proximal term adopted by FedProx, which constrains the global model's evolution in the parameter space, our approach can be viewed as a way to ensure that the new global model is not too far from the previous one in the output space.

The remainder of this section discusses in detail our distillation approach and introduces the Flashback algorithm.

### 4.1 LABEL COUNT DYNAMIC DISTILLATION

In the standard knowledge distillation, all logits are treated equally since it is assumed that the teacher model has been trained on a balanced dataset. Owing to the heterogeneity of data distribution in local datasets, this assumption does not hold in FL. As a result, we cannot directly treat the current global model nor the local model updates as equally reliable teachers across all classes. Instead, we propose to weight the logits by using the label count as an approximation of the per-class knowledge within a model. Here the label count refers to the occurrences of each class in the training data.

We now revisit the distillation loss Eq. (1) and transform the scalar $\alpha$ to a matrix form that is automatically tuned according to the label count and used directly within the KL divergence loss. We consider a single student model $F_{w_s}$ with weights $w_s$ and a set $\mathbb{T}$ of $K$ teacher models; the $i$-th teacher model is denoted as $F_{w_i}$ with weights $w_i$.

Let $\boldsymbol{\nu} \in \mathbb{R}^C$ be the label count vector of the student model, where $\nu^c$ is the occurrences of class $c$ in the dataset. Similarly, let $\boldsymbol{\mu}_i \in \mathbb{R}^C$ be the label count vector of the $i$-th teacher model.

The dynamic $\boldsymbol{\alpha} \in \mathbb{R}^{K \times C}$ is defined as $[\boldsymbol{\alpha}_1^\mathsf{T}, \ldots, \boldsymbol{\alpha}_K^\mathsf{T}]$, with $\boldsymbol{\alpha}_i = \frac{\boldsymbol{\mu}_i}{\boldsymbol{\nu} + \sum_k \boldsymbol{\mu}_k}$.

Then, we embed $\boldsymbol{\alpha}$ directly in the KL divergence loss ($\mathcal{L}_{\text{KL}}$ Eq. (1)) as follows:

$$\mathcal{L}_{\text{dKL}}(\boldsymbol{p}, \boldsymbol{q}; \boldsymbol{\alpha}_i) = \sum_{c=1}^{C} \alpha_i^c \cdot p^c \log \left( \frac{p^c}{q^c} \right)$$

Finally, the dynamic knowledge distillation loss ($\mathcal{L}_{\text{dKD}}$) is:

$$\mathcal{L}_{\text{dKD}}((x, y); w_s, \mathbb{T}, \boldsymbol{\alpha}) = \mathcal{L}_{\text{CE}}(F_{w_s}(x), y) + \sum_{w_i \in \mathbb{T}} \mathcal{L}_{\text{dKL}}(F_{w_i}(x), F_{w_s}(x); \boldsymbol{\alpha}_i) \tag{3}$$

The dynamic $\boldsymbol{\alpha}$ will weigh the divergence between the logits of different classes making the student model focus more on learning from the teacher's strengths while being cautious of its weaknesses. This is of great importance in FL because of the data heterogeneity problem.

### 4.2 FLASHBACK ALGORITHM

Flashback is detailed in Algorithm 1.

Note that to apply the dynamic distillation loss Eq. (3), we require to obtain the student's and teachers' label count vectors. While the label count of local models can be easily obtained (from the class frequency of local datasets), the label count of the global model is not readily available. We construct $\boldsymbol{\pi}$, the global model's label count, as follows. Let $r_k$ denote the number of rounds that client $k$ has participated in. For every client $k$ that participates at round $t$, Flashback adds (Line 27) a fraction $\gamma \in (0, 1]$ of $k$'s label count ($\boldsymbol{\mu}_k$) to $\boldsymbol{\pi}$, unless $\gamma r_k > 1$, in which case $\boldsymbol{\pi}$ is not updated based on $k$'s label count. Progressively building the global label count is crucial for ensuring a balanced distillation weight within the Eq. (3) loss function during the local update. This gradual

---

**Algorithm 1:** Flashback algorithm.

---

**Input:** Initial global model $w_0$, number of rounds $T$, fraction of clients $C$, minibatch size $B$, number of
local epochs $E$, number of server epochs $E_s$, learning rate $\eta$

**Output:** Global model $w_T$

1   $\boldsymbol{\pi} = \mathbf{0}$ ;                  // Global model's label count vector

2   **for** $t \leftarrow 1$ **to** $T$ **do**

3      $\mathbb{S}_t \leftarrow$ (Set of randomly selected $C \cdot N$ clients);

4      **for** *client* $k \in \mathbb{S}_t$ **in parallel do**

5          $w_{k,t} \leftarrow w_{t-1}$ ;               // Initialize local model with current global model

6          $B_k \leftarrow$ (Split local dataset into batches of size $B$);

7          Compute $\boldsymbol{\alpha}$ with $\boldsymbol{\nu}$ as the local label count and a single teacher $\boldsymbol{\mu} \leftarrow \boldsymbol{\pi}$;

8          **for** $e \leftarrow 1$ **to** $E$ **do**

9             **for** *batch* $b \in B_k$ **do**

10               $w_{k,t} \leftarrow w_{k,t} - \eta \cdot \nabla_{w_{k,t}} \mathcal{L}_{\mathrm{dKD}}(b; w_{k,t}, \{w_{t-1}\}, \boldsymbol{\alpha})$ ;      // Use $\mathcal{L}_{\mathrm{CE}}$ when $t = 1$

11            **end**

12          **end**

13      **end**

14      $m_t \leftarrow \sum_{k \in \mathbb{S}_t} n_k$ ;        // Total data points in this round ($n_k$ is the number of data points at $k$)

15      $w_t \leftarrow \sum_{k \in \mathbb{S}_t} \frac{n_k}{m_t} w_{k,t}$ ;             // Average to obtain the new global model

16      $B_s \leftarrow$ (Split the public dataset into batches of size $B$);

17      $\mathbb{T} \leftarrow \begin{cases} \{w_{k,t} \mid k \in \mathbb{S}_t\} \text{ if } t = 1 \\ \{w_{k,t} \mid k \in \mathbb{S}_t\} \cup \{w_{t-1}\} \text{ otherwise} \end{cases}$ ;

18      Compute $\boldsymbol{\alpha}$ with $\boldsymbol{\nu} \leftarrow \boldsymbol{\pi}$ and $\boldsymbol{\mu}_i$ as the label count $\forall w_i \in \mathbb{T}$ ;      // If $t > 1$, $w_{t-1}$ has $\boldsymbol{\mu}_i \equiv \boldsymbol{\pi}$

19      **for** $e \leftarrow 1$ **to** $E_s$ **do**

20          **for** *batch* $b \in B_s$ **do**

21             $w_t \leftarrow w_t - \eta \cdot \nabla_{w_t} \mathcal{L}_{\mathrm{dKD}}(b; w_t, \mathbb{T}, \boldsymbol{\alpha})$ ;        // Update global model

22          **end**

23      **end**

24      $r_k \leftarrow$ (Increment $r_k$ for every client $k \in \mathbb{S}_t$);

25      **for** *client* $k \in \mathbb{S}_t$ **do**

26          **if** $\gamma r_k \leq 1$ **then**

27             $\boldsymbol{\pi} \leftarrow \boldsymbol{\pi} + \gamma \boldsymbol{\mu}_k$;

28          **end**

29      **end**

30 **end**

---

integration reflects the evolving confidence in the global model, safeguarding against overwhelming the client models with a disproportionate weight that could potentially distort the learning process.

## 5   EXPERIMENTS & RESULTS

We outline and analyze our experimental findings to investigate whether mitigating forgetting successfully addresses the issues of slow and unstable convergence observed in the initial problem laid out in § 1. The experimental results stem from six settings with three datasets: two each on CIFAR10 and CINIC10, where heterogeneous data partitions are created using Dirichlet distribution with $\beta = 0.1$ and $\beta = 0.5$, and two on FEMNIST with 500 and 3,432 clients, following the natural heterogeneity of the dataset. We use the same neural network architecture that is used in Lee et al. (2021); McMahan et al. (2017), which is a 2-layer Convolutional Neural Network (CNN). Summaries of the datasets, partitions and more details on the experimental setup, as well as additional results are reported in Appendix B.

We compare Flashback against several baseline methods, namely: 1) FedAvg, 2) FedDF (Lin et al., 2020), 3) FedNTD (Lee et al., 2021), 4) FedProx (Li et al., 2020), 5) FedReg (Xu et al., 2022), 6) MOON (Li et al., 2021b). It is noteworthy that both FedNTD and FedReg target forgetting in FL (discussed further in § 6).

We explore four critical dimensions, encompassing: 1) Round Efficiency which is evaluated as the number of rounds each method takes to reach the target accuracy; 2) Convergence Behavior in which

| Method | CIFAR10 $\beta = 0.1, A = 0.5$ | | | CIFAR10 $\beta = 0.5, A = 0.59$ | | | CINIC10 $\beta = 0.1, A = 0.46$ | | | CINIC10 $\beta = 0.5, A = 0.48$ | | | FEMNIST $N = 500, A = 0.72$ | | | FEMNIST $N = 3,432, A = 0.73$ | | |
|---|---|---|---|---|---|---|---|---|---|---|---|---|---|---|---|---|---|---|
| | $A_{0.75}$ | $A_{0.9}$ | $A_1$ | $A_{0.75}$ | $A_{0.9}$ | $A_1$ | $A_{0.75}$ | $A_{0.9}$ | $A_1$ | $A_{0.75}$ | $A_{0.9}$ | $A_1$ | $A_{0.75}$ | $A_{0.9}$ | $A_1$ | $A_{0.75}$ | $A_{0.9}$ | $A_1$ |
| FedAvg | 112 | 190 | 445 | 28 | 58 | 96 | 269 | 490 | - | 27 | 57 | 135 | 61 | 103 | 213 | 80 | 133 | 265 |
| FedDF | 105 | - | - | 28 | 106 | 139 | 23 | - | - | 18 | 57 | - | - | - | | - | | |
| FedNTD | 71 | 141 | 302 | 28 | 96 | 133 | 41 | 160 | 224 | 24 | 59 | 126 | 177 | 361 | - | 244 | - | - |
| FedProx | 141 | 247 | 441 | 47 | 105 | 161 | 269 | - | - | 43 | 100 | - | 170 | 234 | 442 | 240 | 354 | - |
| FedReg | 126 | 264 | - | 92 | 158 | - | 236 | 312 | - | 100 | - | - | - | - | | - | | |
| MOON | 190 | - | - | 118 | 178 | - | - | - | - | 82 | 169 | - | - | | | - | | |
| Flashback | **11** | **28** | **49** | **4** | **12** | **30** | **5** | **9** | **17** | **5** | **11** | **44** | **2** | **5** | **16** | **2** | **6** | **21** |

Table 1: Number of rounds to reach accuracy $A_x = A \cdot x$ where $A$ is the target accuracy and $x$ is a fraction of it. The target accuracy is set at 95% of $A_{\max}$, the highest obtained accuracy by any of the algorithms. $\beta$ is the Dirichlet distribution parameter, and $N$ is the total number of clients. A dash (-) indicates that a method failed to reach the accuracy.

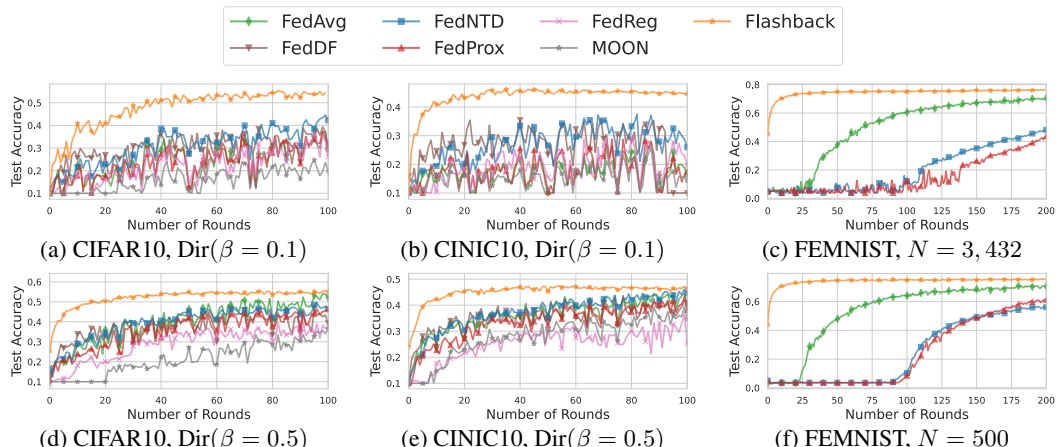

Figure 3: Test accuracy (y-axis) over rounds (x-axis) for different algorithms and datasets.

we observe the rate at which an algorithm can improve its global model accuracy and the level of oscillations in accuracy throughout the rounds; 3) Global & Local Forgetting in which we measure how much forgetting happens for each method. 4) Knowledge Absorption in which we dissect the accuracy metric at the class level and observe the learning behavior.

**Convergence Behavior & Round Efficiency.** We report accuracy over the rounds in Fig. 3 as well as rounds to reach target accuracy in Table 1. Flashback demonstrates faster and more stable convergence compared to the other baselines across all the experiments and the round efficiency is improved by up to $29.5\times$, $54.4\times$, and $88.5\times$ in CIFAR10, CINIC10, and FEMNIST, respectively. This indicates that mitigating forgetting addresses the slow convergence and oscillation problem discussed in § 1. In CIFAR10 and CINIC10, FedNTD performs the best out of the other baselines; moreover, when data is less heterogeneous (Dir($\beta = 0.5$)), FedAvg is on bar with the other baselines. In FEMNIST, 3 of the baselines fail to converge (FedReg, FedDF, MOON), and most notably FedAvg performs the best among the baselines; though Flashback is clearly superior with nearly one order of magnitude fewer rounds.

**Global & Local Forgetting.** We measure the global and local forgetting scores over rounds as evaluated by Eq. (2). Fig. 4 shows their distributions. We observe that Flashback achieves the lowest global forgetting score compared to other methods while sometimes it achieves a worse local forgetting score than other methods (this improves when data is less heterogeneous, i.e., Dir($\beta = 0.5$); see Fig. 14). This is an interesting result, as it shows that mitigating global forgetting is correlated with the problem of slow and unstable convergence. While mitigating local forgetting is not. In fact, local forgetting might be necessary to obtain new knowledge from the local update as long as the local models' knowledge can be aggregated properly and knowledge from the previous round global model $w_{t-1}$ is not forgotten in $w_t$, which is the case for Flashback. Therefore, global forgetting is a more critical problem, and hence the aggregation step is more important to enhance.

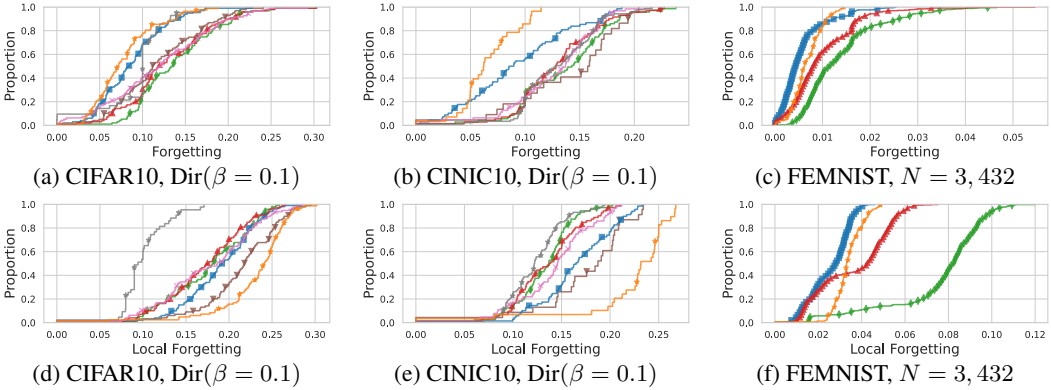

Figure 4: CDF of global forgetting $\mathcal{F}_t$ (above) and local forgetting $\mathcal{F}_{k,t}$ (below) over the rounds for different algorithms and datasets. Legend is the same as in Fig. 3.

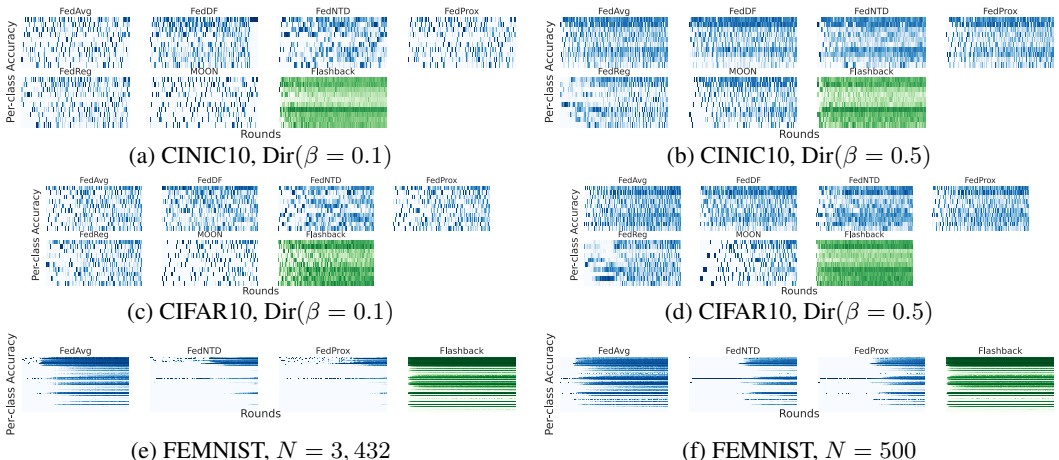

Figure 5: Global model per-class accuracy over rounds.

**Knowledge Absorption.** We show the learning behavior of all the algorithms via the per-class accuracy over the rounds as heatmaps shown in Fig. 5. Flashback shows more stability and fewer light-shaded boxes, visually showing its robustness to forgetting in comparison with the baselines.

## 5.1 DISCUSSION

**Public dataset assumption.** As with prior work Huang et al. (2022); Zhang et al. (2021); Lin et al. (2020); Cheng et al. (2021); Li et al. (2020), Flashback assumes availability of a public dataset. This is to perform the distillation at the aggregation step of the algorithm. However, as detailed in Appendix B.1, the requirement is minimal, that is, in the case of CIFAR10 and CINIC10, the public dataset, comprising just 1,250 and 4,500 data points, respectively, was smaller in size than some of the clients' datasets. In the future, we aim to evaluate diverse public dataset distributions.

**Additional computational cost.** Flashback performs knowledge distillation during the local update, which has smaller cost than doing an additional local epoch. That is because the global model logits need to be computed once, using forward passes only. This cost is similar to FedNTD and smaller than MOON, which does 2 additional forward passes.

## 6 RELATED WORK

**Federated learning.** FL is commonly viewed as a ML paradigm wherein a server distributes the training process on a set of decentralized participants that train a shared global model using local

datasets that are never shared (Konečný et al., 2015; Shokri & Shmatikov, 2015; Konečný et al., 2016; Konečný, 2017; Li et al., 2020; McMahan et al., 2017; Kairouz et al., 2019). FL has been used to enhance prediction quality for virtual keyboards among other applications (Bonawitz et al., 2019; Yang et al., 2018). A number of FL frameworks have facilitated research in this area (Caldas et al., 2019; OpenMined, 2020; tensorflow.org, 2020; Abdelmoniem et al., 2023).

**Heterogeneity in FL.** A key challenge in FL systems is uncertainties stemming from learner, system, and data heterogeneity. The non-IID distributions of Learners' data can significantly slow down convergence (McMahan et al., 2017; Kairouz et al., 2019) and several algorithms are proposed as means of mitigation (Wang et al., 2020b; Karimireddy et al., 2020; Li et al., 2020; 2021a).

**Forgetting in FL** is an under-studied area that poses significant challenges, leading to slow model convergence and loss of crucial knowledge acquired during the learning process (Chaudhry et al., 2018; Dupuy et al., 2023). There have been some notable attempts to mitigate the impact of forgetting on the learning process (Lee et al., 2021; Xu et al., 2022).

**FedReg** (Xu et al., 2022) addresses the issue of slow convergence in FL, asserting it to be a result of forgetting at the local update phase. They demonstrate this by comparing the loss of the global model $w_{t-1}$ on specific client data points with the averaged loss of updated clients' models $\{w_{t,k} \mid k \in \mathbb{S}_t\}$ on the same data points, highlighting a significant increase in the average loss, indicative of forgetting. However, in our work, we propose a systematic way of measuring forgetting using a metric designed to capture it. Furthermore, we show that forgetting doesn't only occur in the local update, but it also happens at the aggregation step. FedReg proposes to generate fake data that carries the previously attained knowledge. During the local update, Fast Gradient Sign Method (Goodfellow et al., 2014) is used to generate these data using the global model $w_{t-1}$ and the client data. Then, the loss of the generated data is used to regularize the local update. While FedReg employs regularization using synthetic data during local updates, our work, Flashback, leverages dynamic distillation to ensure knowledge retention at both local updates and aggregation steps.

**FedNTD** (Lee et al., 2021) makes a connection between CL and FL, suggesting that forgetting happens in FL as well. Similarly to FedReg, their analysis shows that forgetting happens at the local update, where global knowledge that lies outside of the local distribution of the client is susceptible to forgetting. To address this, they propose to use a new variant of distillation Eq. (1) named Not-True Distillation (NTD), that masks the ground-truth class logits in the KL divergence as $\mathcal{L}_{\mathrm{KL}}(\boldsymbol{p}, \boldsymbol{q}) = \sum_{i=c,c \neq y}^{C} p^c \log(\frac{p^c}{q^c})$, where $y$ is the ground-truth class. NTD is used at the local update, while all the other steps in the algorithm remain the same as FedAvg. FedNTD aims to preserve global knowledge during the local update.

Both FedReg and FedNTD diagnose the issue of forgetting primarily within the realm of local updates, asserting that this stage risks losing valuable global knowledge. Consequently, both works present innovative solutions specifically tailored to counteract this local update forgetting. However, their perspective overlooks a pivotal aspect of the forgetting problem: the occurrence of forgetting during the aggregation step. As we delve into in § 3, this oversight in recognizing and addressing forgetting during aggregation has repercussions on the later local updates. In contrast, Flashback takes a holistic approach, targeting forgetting comprehensively across both the local updates and the aggregation phase, leading to faster convergence.

## 7  CONCLUSION

We explored the phenomenon of forgetting in FL. Our investigation revealed that forgetting occurs during both local and global update phases of FL algorithms. We presented Flashback, a novel FL algorithm explicitly designed to counteract forgetting by employing dynamic knowledge distillation. Our approach leverages data label counts as a proxy for knowledge, ensuring a more targeted and effective forgetting mitigation. Our empirical results showed Flashback's efficacy in mitigating global forgetting, thereby supporting the hypothesis that the observed slow and unstable convergence in FL algorithms is closely linked to global forgetting. This result underlines the importance of addressing forgetting, paving the way for the advancement of more robust and efficient FL algorithms.

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
