# A  LABEL COUNT MOTIVATION

As established in § 3, the client local model can forget, and contain only the knowledge present in the private data. Moreover, even the global model can be imperfect, for two reasons: i) As we established before the global model is susceptible to forgetting in the aggregation step. ii) Assuming no forgetting in the aggregation step, the knowledge contained in the clients who participated so far, might not represent all the available knowledge, especially in the early rounds. Overall, both local models and the global model can be imperfect. Therefore, the logits of all the different classes can not be treated equally. To this end, devising an approach to estimate the knowledge of the different models in the FL process is crucial. We propose to use the label count as an approximation of the knowledge within a model. In machine learning, a model's knowledge is fundamentally tied to the data it has been exposed to. If certain classes have higher representation (or label counts) in the training data, it's intuitive that the model would have more opportunities to learn the distinguishing features of such classes. Conversely, underrepresented classes might not offer the model sufficient exposure to learn their nuances effectively. Our extensive experiments with FL training and separate local training have consistently shown that the per-class performance on the test set correlates highly with the label counts in the training data. In scenarios where certain classes were more abundant, the model demonstrated higher proficiency in predicting those classes on the test set. For instance, in Fig. 6 the label count reflected the performance of the client's local model on the test. Such a consistent observation strengthens the claim that the label count serves as a tangible measure of the model's knowledge.

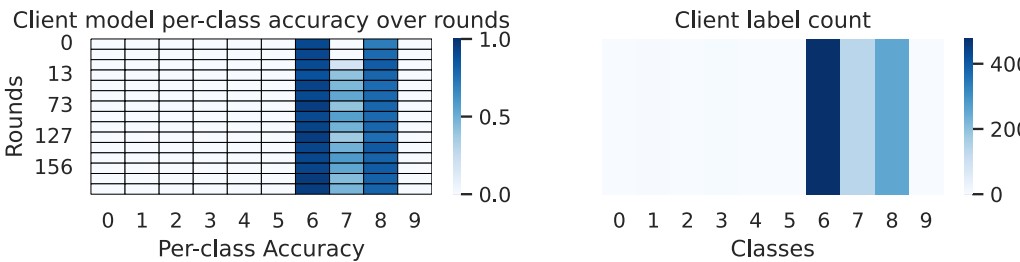

Figure 6: The first part of the figure shows the per-class accuracy of a client model on all the rounds where it participated. The second part shows the data distribution of that client.

# B  EXPERIMENTS DETAILS

## B.1  DATASETS

In this section, we provide an overview of the datasets used, the data split, and the specific experimental setups. For each dataset, we perform two sets of experiments to analyze the effects of data heterogeneity on the algorithms' performance. The datasets used are CIFAR10, CINIC10, and FEMNIST.

**CIFAR10** (Krizhevsky, 2009). A famous vision dataset that includes 50k training images and 10k testing images. We emulate a realistic, heterogeneous data distribution by using a Dirichlet distribution with parameters $\beta = 0.1$ and $\beta = 0.5$. A lower $\beta$ value of 0.1 is chosen to simulate a more heterogeneous, and challenging data distribution. A 2.5% random sample of the training set creates a public dataset, further divided into training and validation sets. The remaining 97.5% is distributed among 100 clients, with each client's data being split into training (90%) and validation (10%) subsets.

**CINIC10** (Darlow et al., 2018). An extension of CIFAR10, this dataset comprises 90k training, 90k validation, and 90k test images. We merge the training and validation sets and adopt a similar approach as with CIFAR10, taking out 2.5% of the data to be the public dataset, then employing $\beta$ values of 0.1 and 0.5 in the Dirichlet distribution to split the 97.5% remaining data into 200 clients, with each client's data further divided into training (90%) and validation (10%) sets.

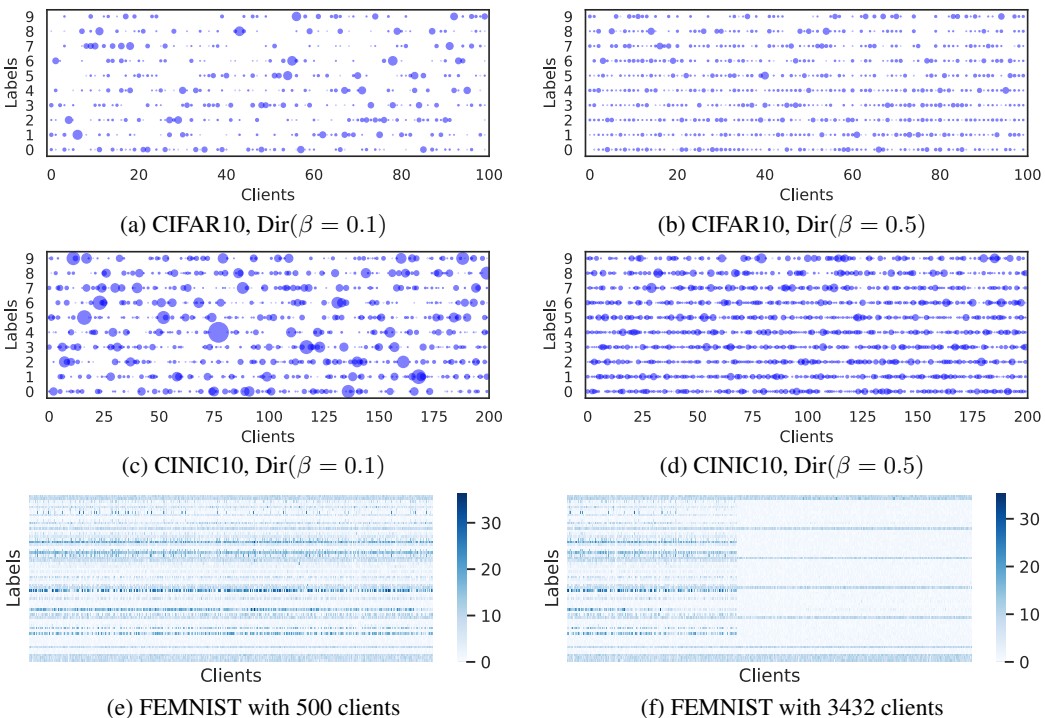

Figure 7: Clients data distribution. The x-axis is the clients and the y-axis is the labels.

**FEMNIST** (Caldas et al., 2019). This federated learning dataset is based on extended MNIST with natural heterogeneity, where each writer is considered a client. From the 3597 total writers, those with less than 50 samples are excluded. We randomly selected 150 writers to form a public dataset. The remaining 3432 writers' data is divided into train (approx. 70%), validation (approx. 15%), and test (approx. 15%) sets. The collective test sets from all writers form the overall test set. Two experimental setups are used: 1) The first 500 clients are selected, with 16 clients randomly chosen for participation in each round; and 2) All 3432 clients are used, with 32 clients randomly chosen for participation in each round.

For CIFAR10 and CINIC10, we perform two sets of experiments with $\beta$ values of 0.1 and 0.5 with the client participation value of 10. While for FEMNIST we vary $N$ and the client participation, since the data split is realistic by nature. In all cases, the training data distribution among clients is illustrated in Fig. 7.

### B.2 BASELINES & HYPERPARAMETERS

We evaluate the following algorithms as baselines: 1. FedAvg (McMahan et al., 2017); 2. FedDF (Lin et al., 2020); 3. FedNTD (Lee et al., 2021); 4. FedProx (Li et al., 2020); 5. FedReg (Xu et al., 2022); and 6. MOON (Li et al., 2021b) .

Both FedNTD and FedReg target forgetting in FL (discussed in § 6). We use the same neural network architecture that is used in Lee et al. (2021); McMahan et al. (2017), which is a 2-layer CNN. Moreover, for the optimizer, learning rate, and model we follow Lee et al. (2021); McMahan et al. (2017), and when a baseline has different hyperparameters we use their proposed values. For example, in FedDF the number of local epochs is set to 40, while in the other baselines, it is set to 5 epochs. As for Flashback-specific hyperparameters, we set the number of epochs $E_s = 2$ for the server update, while the local epochs $E = 20$; we set the label count fraction $\gamma = 0.025$, i.e., we add 2.5% of the client label count each time it participates. As for distillation-specific hyperparameters, we have one fewer hyperparameter since $\alpha$ is computed automatically, and for temperature, we use the standard $T = 3$.

To ensure the reliability of our results, we additionally trained a central model on both the entire dataset and the public dataset, as well as FedAvg with an IID data split, ensuring consistency with the previously defined models and hyperparameters. For the CIFAR10 dataset, central training on the entire dataset yielded a test accuracy of 70%, while the FedAvg with an IID split achieved a test accuracy of 66.4%. Central training on the public dataset for CIFAR10 reported a test accuracy of 40.06%. For the CINIC10 dataset, central training on the entire and public datasets reported test accuracies of 55% and 39.2% respectively, whereas FedAvg with an IID split also reported a test accuracy of 55%.

## C ADDITIONAL RESULTS

In this section, we show supplementary results. We present plots depicting the test and validation loss over rounds in Fig. 8 and Fig. 9, respectively. We present local and global forgetting over rounds in Fig. 13. Then, we present the average forgetting scores of the evaluated methods in Table 2. We then present the per-class forgetting for all methods in Fig. 10. We show also the dynamics of the global and local forgetting over rounds in Fig. 11 and Fig. 12, respectively. Finally, the CDF of the global and local forgetting for various methods in the lower data heterogeneity scenario is depicted in Fig. 13 and Fig. 14, respectively. Unless otherwise noted, the legend is common across figures and it is as in Fig. 8.

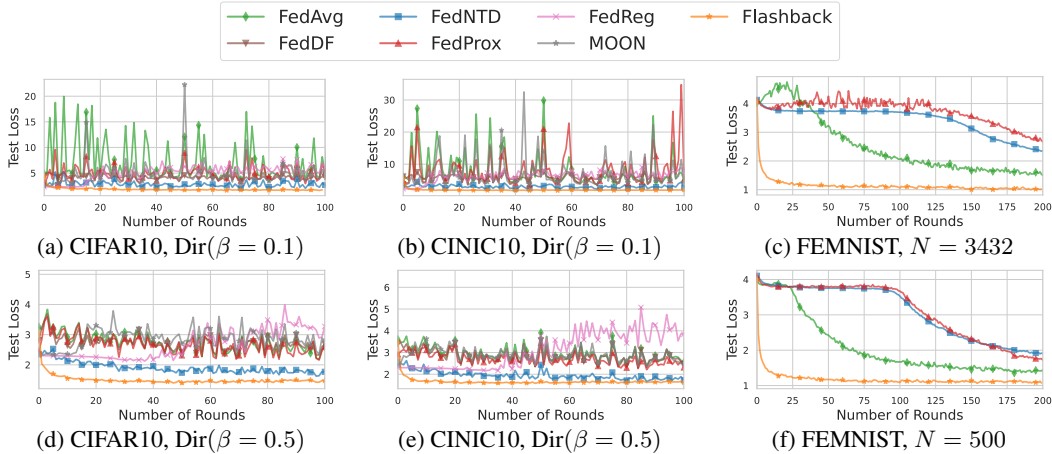

Figure 8: Average clients test loss (y-axis) over rounds (x-axis) for different algorithms and datasets

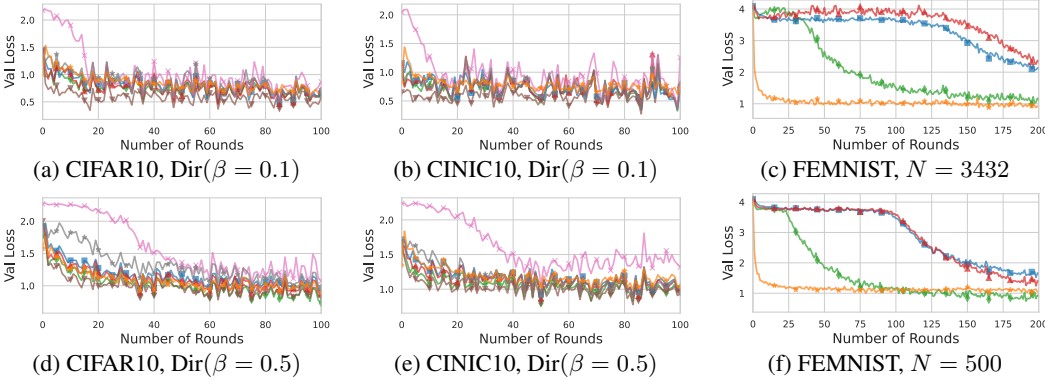

Figure 9: Average clients validation loss (y-axis) over rounds (x-axis) for different algorithms and datasets. The validation set reflects the local client distribution.

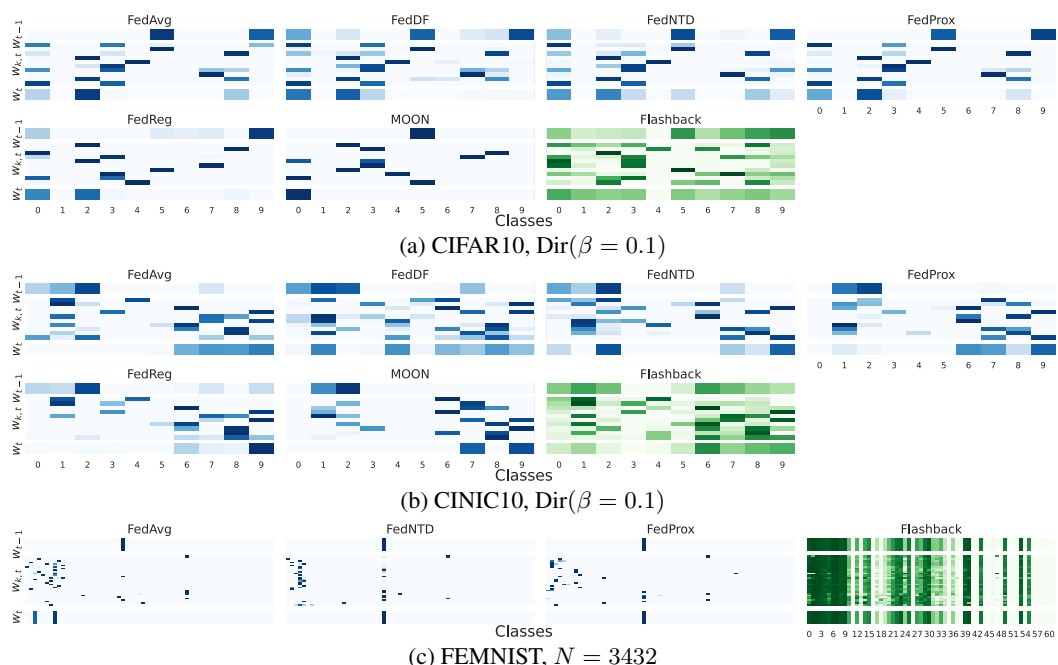

Figure 10: Local (client) & Global Forgetting in all the baselines. The first row represents the global model per-class test accuracy at round $t-1$; then the rows in the middle are all the clients that participated in round $t$, and finally in the last row the global model at the end of round $t$. Local forgetting happens when clients at round $t$ lose the knowledge that the global model had at round $t-1$. The global forgetting happens when the global model at round $t$ loses the knowledge that in the clients' models at round $t$.

| Method | CIFAR10 $\beta=0.1$ Global | Local | CIFAR10 $\beta=0.5$ Global | Local | CINIC10 $\beta=0.1$ Global | Local | CINIC10 $\beta=0.5$ Global | Local | FEMNIST $N=500$ Global | Local | FEMNIST $N=3432$ Global | Local |
|---|---|---|---|---|---|---|---|---|---|---|---|---|
| FedAvg | 0.139 | 0.176 | 0.142 | 0.215 | 0.139 | 0.136 | 0.118 | 0.202 | 0.018 | 0.065 | 0.014 | 0.077 |
| FedDF | 0.119 | 0.213 | 0.116 | 0.201 | 0.144 | 0.197 | 0.103 | 0.192 | - | | - | |
| FedNTD | 0.087 | 0.189 | 0.076 | 0.179 | 0.094 | 0.170 | 0.063 | 0.170 | 0.004 | 0.019 | 0.006 | 0.026 |
| FedProx | 0.130 | 0.171 | 0.134 | 0.202 | 0.130 | 0.136 | 0.113 | 0.201 | 0.009 | 0.028 | 0.010 | 0.037 |
| FedReg | 0.120 | 0.176 | 0.088 | 0.169 | 0.128 | 0.146 | 0.149 | 0.196 | - | | - | |
| MOON | 0.117 | 0.122 | 0.124 | 0.148 | 0.130 | 0.122 | 0.138 | 0.187 | - | | - | |
| Flashback | 0.078 | 0.231 | 0.032 | 0.130 | 0.054 | 0.226 | 0.052 | 0.148 | 0.007 | 0.045 | 0.007 | 0.034 |

Table 2: Average forgetting scores for different methods across datasets.

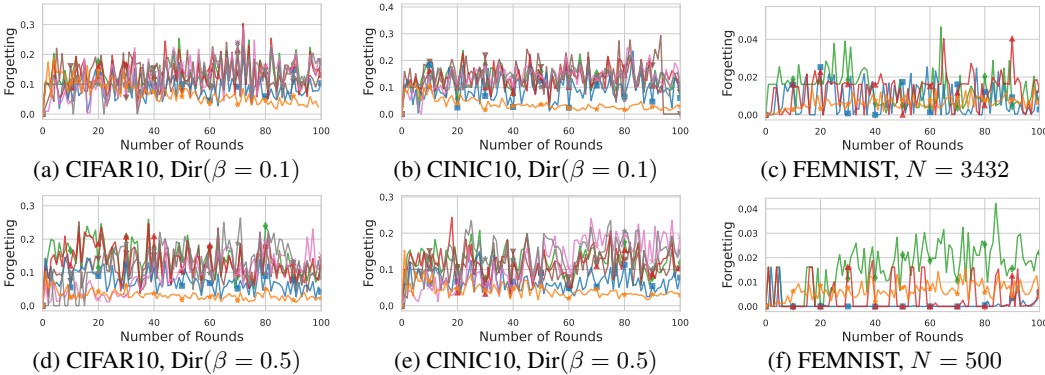

Figure 11: Global forgetting score (y-axis) over rounds (x-axis) for different algorithms and datasets

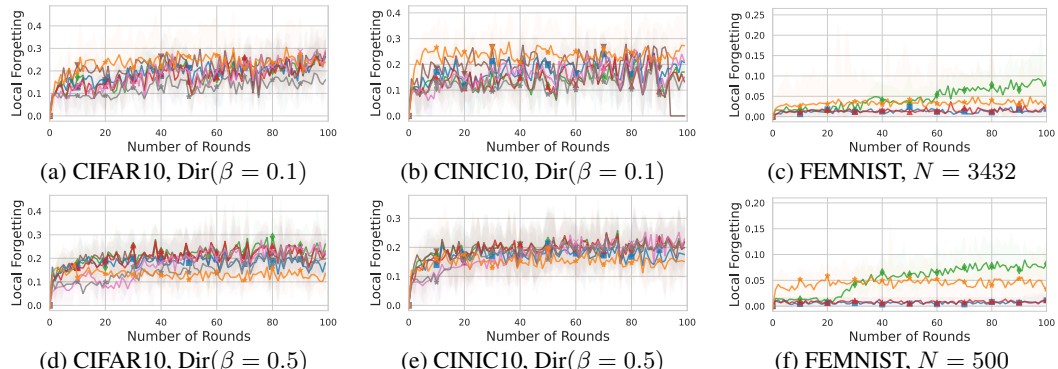

Figure 12: Average local forgetting score (y-axis) over rounds (x-axis) for different algorithms and datasets

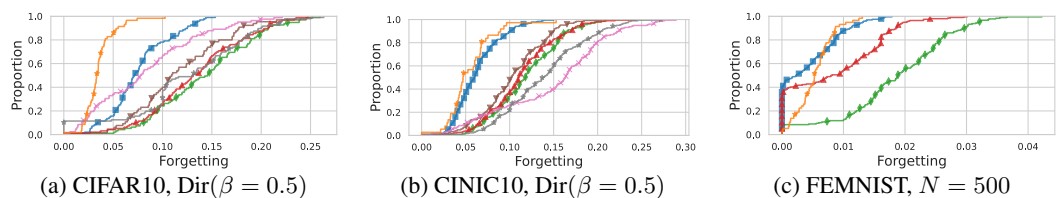

Figure 13: CDF of global forgetting score over the rounds for different algorithms and datasets

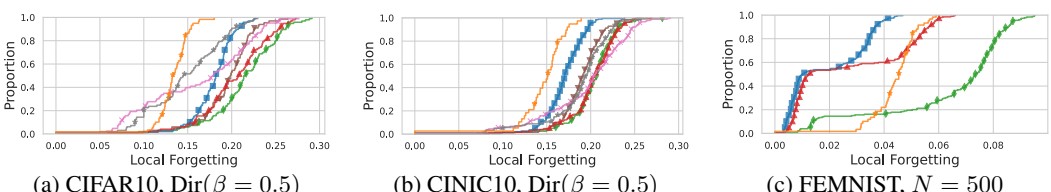

Figure 14: CDF of local forgetting score over the rounds for different algorithms and datasets