# OpenReview forum: "Flashback: Understanding and Mitigating Forgetting in Federated Learning"
_ICLR.cc/2024/Conference — Submitted to ICLR 2024_

### Official Review · Reviewer_LiW6 · 2023-10-29

**Soundness:** 3 good
**Presentation:** 3 good
**Contribution:** 3 good
**Rating:** 5
**Confidence:** 4

**Summary:**

This paper investigates the catastrophic forgetting issue in FL that occurs in both local training and server aggregation. It provides empirical analysis and insights into the forgetting issue and introduces a new method to mitigate this forgetting issue. The proposed method achieves much better convergence than the compared counterparts.

**Strengths:**

- The forgetting issue in FL is important and the analysis and the introduced method are technically sound.
- The proposed method achieves much better convergence than the compared counterparts.
- The paper is generally well-written and easy to follow.

**Weaknesses:**

- Some experimental details seem to be missing. e.g., what is the public dataset that is used for experiments on CIFAR, CINIC, and FEMNIST?
- The comparison with other methods may not be fair as the proposed method leverages a shared public dataset in the server while compared methods may not use it. Some papers on FL and KD also use a public dataset.  e.g., [1][2].
    - [1] Ensemble distillation for robust model fusion in federated learning. NeurIPS’20
    - [2] Performance optimization of federated person re-identification via benchmark analysis. ACMMM’20
- A straightforward baseline to consider is fine-tuning with the public dataset in the server, using soft labels from clients or ground truth labels. It would provide more insights into the significance of the proposed method. The reviewer would consider raising the rating if some of the concerns can be addressed.

**Questions:**

- What is the impact of different selection choices of public datasets? Would the method still work if the data distribution of the public dataset is different from the client’s data distribution?
- What is the backbone used to train CIFAR and CINIC datasets? Is the method robust across different backbones?

---

> ### Author Response · Authors · 2023-11-17
>
> We sincerely thank the reviewer for their time and effort, and appreciate all of their feedback!
>
>
> ## Weaknesses
>
> - Some experimental details seem to be missing. e.g., what is the public dataset that is used for experiments on CIFAR, CINIC, and FEMNIST?
>     - We kindly refer the reviewer to section 5.1 and B.1 in the appendix.
> - The comparison with other methods may not be fair as the proposed method leverages a shared public dataset in the server while compared methods may not use it. Some papers on FL and KD also use a public dataset. e.g., [1][2].
>     - This is a valid concern. Indeed, for this reason we compare with [1]. Moreover, FedDF [1] uses the all of CIFAR100 as the public dataset, when experimenting with CIFAR10, while we only use 2.5% of CIFAR10 as the public dataset, and use the remaining 97.5% for the clients. This is a much smaller public dataset.
> - A straightforward baseline to consider is fine-tuning with the public dataset in the server, using soft labels from clients or ground truth labels. It would provide more insights into the significance of the proposed method. The reviewer would consider raising the rating if some of the concerns can be addressed.
>     - This a very good suggestion, we will add such an experiment in the revised manuscript.
>
> ## Questions
>
> - What is the impact of different selection choices of public datasets? Would the method still work if the data distribution of the public dataset is different from the client’s data distribution?
>     - We experimented with a very small public dataset, we would report on additional experiments to answer this question in the revised manuscript.
> - What is the backbone used to train CIFAR and CINIC datasets? Is the method robust across different backbones?
>     - We use the same backbone across all the datasets (CIFAR, CINIC, and FEMNIST), and we expect the method to be general across different model architectures. We will do additional experiments to verify the robustness of our method across models.

---

> > ### Comment · Reviewer_LiW6 · 2023-11-19
> > **Response to Authors**
> >
> > Thank you for addressing some of the reviewer's concerns.
> >
> > Apart from the replies that promise to add new experiments, the reviewer still has a concern about the comparison with FedDF. Although FedDF uses the whole CIFAR100 as a public dataset, the proposed method uses a small fraction of CIFAR10, which is the same dataset as the testing data. It would be more convincing if FedDF also used the CIFAR10 dataset as the public dataset.

---

> ### Author Response · Authors · 2023-11-19
>
> Thanks for the reply!
>
> Apologies for the confusion I caused by my comment. [1] in their own experiments, they used CIFAR100 as the public dataset, when training on CIFAR10. In our experiments, we used the same public dataset for all the methods that use a public dataset. So for both Flashback and FedDF [1] we used the exact same small fraction of CIFAR10 as the public dataset.

---

### Official Review · Reviewer_VJec · 2023-10-31

**Soundness:** 2 fair
**Presentation:** 2 fair
**Contribution:** 2 fair
**Rating:** 5
**Confidence:** 3

**Summary:**

The paper studies the problem of forgetting in FL. Specifically, they show that several standard federated optimization methods can fail in high-heterogeneity settings due to local and global forgetting, which respectively occur during client training and server aggregation. To address this, the authors use distillation at both the server and client during FL. At the server, they distill an ensemble of models from client fine-tuning and the previous server round into a new server model. At the client, they distill the initial (l.e. server) model into the fine-tuned model. The distillation additionally weights the logits based on (aggregated) client label counts.

**Strengths:**

The paper makes an interesting point about how prior works focus on making either the global or local step robust but fail to consider both.

The method outperforms a variety of baselines which use regularization / distillation.

**Weaknesses:**

Please closely examine the claim in Discussion 5.1 about related work. Li et al. 2020 (FedProx) makes no assumptions about public server-side data.

It would be good if you can include an ablation on using distillation only at the server / clients. The paper claims (page 4, above Fig.2) "Moreover, local forgetting and global forgetting are intertwined, which means addressing the issue at only one of the phases will not be sufficient, since it will happen at the next phase, and therefore have a cascading effect into the same phase at the next round."
I think this statement intuitively makes sense but it could use more support.

More generally, an ablation on various components of Flashback would be helpful. Relative to FedDF there are a lot of things going on, i.e. including the previous round teacher, weighting the logits, and local distillation. Based on the story of the paper I would expect adding local distillation to be the most important factor.

Figure 2 would be more helpful if you use the same initial model for all methods. Also consider only showing one row of methods.

**Questions:**

FedDF reports very high numbers on CIFAR10 (Table 1 in https://proceedings.neurips.cc/paper_files/paper/2020/file/18df51b97ccd68128e994804f3eccc87-Paper.pdf). Was the evaluation of FedDF too limited? Why can they reach up to 75% in those experiments?

---

> ### Author Response · Authors · 2023-11-17
>
> We sincerely thank the reviewer for their time and effort, and appreciate all of their feedback!
>
> ## Weaknesses
>
> - Please closely examine the claim in Discussion 5.1 about related work. Li et al. 2020 (FedProx) makes no assumptions about public server-side data.
>     - Thanks for the observation this is a mistake on our part, indeed FedProx does not assume or use a public dataset, we will fix this typo in the revised manuscript.
> - It would be good if you can include an ablation on using distillation only at the server / clients. The paper claims (page 4, above Fig.2) "Moreover, local forgetting and global forgetting are intertwined, which means addressing the issue at only one of the phases will not be sufficient, since it will happen at the next phase, and therefore have a cascading effect into the same phase at the next round." I think this statement intuitively makes sense but it could use more support.
>     - Indeed we have done further experiments that explore this question, the experimental results do support our statement. We will add this experiment to the revised manuscript.
> - More generally, an ablation on various components of Flashback would be helpful. Relative to FedDF there are a lot of things going on, i.e. including the previous round teacher, weighting the logits, and local distillation. Based on the story of the paper I would expect adding local distillation to be the most important factor.
>     - Yes the combination of doing distillation at both ends of the FL algorithm is an important to addressing forgetting, we will add additional experiments in the revised manuscript that explore the necessity of the additional components of Flashback.
> - Figure 2 would be more helpful if you use the same initial model for all methods. Also, consider only showing one row of methods.
>     - Thanks for the feedback. Indeed, in our methodology we start with the same initial weights, and we deterministically fix the data partitioning and the selected clients for all the methods. We will update the text to reflect this setup.
>
> ## Questions
>
> - FedDF reports very high numbers on CIFAR10 (Table 1 in https://proceedings.neurips.cc/paper_files/paper/2020/file/18df51b97ccd68128e994804f3eccc87-Paper.pdf). Was the evaluation of FedDF too limited? Why can they reach up to 75% in those experiments?
>     - In FedDF they have a different experimental setup, one of the biggest differences is the number of clients in the experiment, they experiment with only 20 clients for table 1.

---

### Official Review · Reviewer_eqJf · 2023-10-31

**Soundness:** 2 fair
**Presentation:** 3 good
**Contribution:** 1 poor
**Rating:** 3
**Confidence:** 4

**Summary:**

The paper targets the problem of data heterogeneity in the federated setting. The authors introduce two sources of performance degradation in non-iid settings: local forgetting and global forgetting. To mitigate the forgettings, they propose FLASHBACK, which employs weighted knowledge distillation on the client and server sides. Clients use the global model as their teacher, and the server uses all the new updates + the last round model as a set of teachers.

**Strengths:**

* The evaluations show the superiority of FLASHBACK.
* The evaluations are comprehensive on different metrics.
* Using the global/local forgettings improves the understanding of the underlying problem in heterogenous FL, and it should be considered in this area as well.

**Weaknesses:**

* The paper assumes that the distribution of the public data is the same as training data (public data is a part of the original dataset). In other words, in the experiments, representative public data is available, which does not usually happen in reality.
* It is unclear if the other baseline methods benefit from the public dataset. Their performance can improve if the server can train on the centralized public dataset as well.
* The algorithm has two parts: local and global KD. An ablation study on each part needs to be included.
* Using KD in the clients and server and forgetting in federated learning is not new. Plenty of previous works, such as [1], use KD in client and server to mitigate forgetting.
* Sharing label information with the server is not privacy-preserving.

[1] Ma, Yuhang, et al. "Continual federated learning based on knowledge distillation." IJCAI 2022.

**Questions:**

* How does your method work on more complex datasets or models?
* How does your paper compare with the federated continual learning papers?
* Please check out the weakness section for the rest of the questions.

---

> ### Author Response · Authors · 2023-11-17
>
> We sincerely thank the reviewer for their time and effort, and appreciate all of their feedback!
>
> ## Weaknesses
>
> - The paper assumes that the distribution of the public data is the same as training data (public data is a part of the original dataset). In other words, in the experiments, representative public data is available, which does not usually happen in reality.
>     - Response: Addressed in a shared response about the public dataset.
> - It is unclear if the other baseline methods benefit from the public dataset. Their performance can improve if the server can train on the centralized public dataset as well.
>     - Response: We include a comparison against FedDF, which uses a public dataset. FedDF has poorer performance compared to some other baselines (see Figure 3). This indicates that using a public dataset doesn’t necessarily give a performance boost.
> - The algorithm has two parts: local and global KD. An ablation study on each part needs to be included.
>     - Response: We did an additional experiment that explored this question, and we found out that, using both is necessary. And doing KD on one side only doesn’t yield a good performance and training stability. This is experiment will be included in the revised manuscript. Check this image for the result: https://ibb.co/sWdp6JQ
>
>
>
> - Using KD in the clients and server and forgetting in federated learning is not new. Plenty of previous works, such as [1], use KD in client and server to mitigate forgetting.
>     - Response: We agree with the reviewer that using KD in FL and forgetting problems are not new. However, the perspective about forgetting and FL and the deeper analysis of forgetting in FL is a new contribution. Moreover, the continual federated learning addresses a different forgetting problem that mainly stems from the change of tasks, exactly like continual learning; however, in our work, we show that forgetting as a stand-alone problem in deep learning, does occur in the federated learning setup. We kindly ask the reviewer to refer to section 3 of the paper, where our analysis of forgetting in FL is new and novel.
>
> ## Questions
>
> - How does your method work on more complex datasets or models?
>     - Response: We do use the standard models and datasets used in the literature. It is an interesting question that we could answer as an additional result but is not necessary to support the validity of Flashback.
> - How does your paper compare with the federated continual learning papers?
>     - Response: In Flashback we do not tackle continual learning at all. However, since federated continual learning does training on one task at a time, that specific training on each individual task can be seen as a standard FL training, therefore, the observations we discussed in our paper regarding forgetting apply during the training of the tasks in federated continual learning. Perhaps if the reviewer has a more specific question, we can address and discuss the comparison of our work and federated continual learning.
> - Please check out the weakness section for the rest of the questions.
>     - Response: Addressed

---

### Official Review · Reviewer_YZyz · 2023-11-07

**Soundness:** 3 good
**Presentation:** 3 good
**Contribution:** 1 poor
**Rating:** 3
**Confidence:** 4

**Summary:**

The paper studies the problem of forgetting in federated learning, particularly in contexts where statistical heterogeneity across clients in high. To do this, the manuscript proposes a new metric to measure forgetting both at the client level (after local updates) and at the the global level (after aggregation at the server), and a method to alleviate this phenomenon by further distilling the knowledge of the local models plus the last round's model. Results are presented on three benchmark datasets.

**Strengths:**

- The problem of forgetting in federated learning is important and timely.
- The paper is presented in such a way that it builds upon simple ideas that are intuitive and easy to follow.
- The use of distillation to mitigate forgetting seems natural given the connections of distillation to predictive churn [1].

[1] Jiang, H., Narasimhan, H., Bahri, D., Cotter, A., & Rostamizadeh, A. (2021, October). Churn Reduction via Distillation. In International Conference on Learning Representations.

**Weaknesses:**

- One of the stated contributions of the paper is to "show how and where forgetting happens in FL". I'm not convinced this question is answered by the manuscript. In particular, only two possible causes are explored: local training and server aggregation. Other possible factors are not considered, e.g., the ordering of the clients, the ordering of the data in the clients [2]. I believe the manuscript should be more specific in this statement or, hopefully, perform a more systematic exploration of what really affects forgetting in FL.
- In the same line, the paper defines, measures and tests local forgetting. Later on, it concludes that some amount of it is necessary for learning. This conclusion is valuable, but this nuance is not reflected in the introduction nor in the motivation of the manuscript.
- There is little discussion of the public data used by the algorithm until Section 5.1. Even then, I am left with questions regarding how it can affect the forgetting behavior. What distribution does this data need to be drawn from? Can it exacerbate forgetting if drawn from the wrong distribution?
- I am surprised that several baselines did not converge for FEMNIST in Table 1. This is a fairly simple benchmark that should achieve good performance with a CNN.

[2] Toneva, M., Sordoni, A., des Combes, R. T., Trischler, A., Bengio, Y., & Gordon, G. J. (2018, September). An Empirical Study of Example Forgetting during Deep Neural Network Learning. In International Conference on Learning Representations.

**Questions:**

- I found Figures 2 and 5 confusing. I'm not sure what the colors refer to, and why Flashback is performing better according to these figures. Please clarify.
- In line with the connections between distillation and algorithmic churn, and with other possible causes of forgetting, future versions of the manuscript would benefit from studying forgetting at the example level (see [2]) for a given test dataset at the server.

---

> ### Author Response · Authors · 2023-11-17
>
> We sincerely thank the reviewer for their time and effort, and appreciate all of their feedback!
>
> ## Weaknesses
>
> - One of the stated contributions of the paper is to "show how and where forgetting happens in FL". I'm not convinced this question is answered by the manuscript. In particular, only two possible causes are explored: local training and server aggregation. Other possible factors are not considered, e.g., the ordering of the clients, the ordering of the data in the clients [2]. I believe the manuscript should be more specific in this statement or, hopefully, perform a more systematic exploration of what really affects forgetting in FL.
>     - *Response*: That is a very interesting point, and can be an additional dimension of exploring forgetting in FL. However, we believe that the first example is implicitly included in the **server forgetting,** where generally given a random order of clients of the rounds forgetting would occur, therefore, constructing a difficult ordering of clients will make the **server forgetting** more severe. This is not to say that this specific example does not need to be studied but to say that our characterization is more general. As an additional supporting material, we will update the manuscript to make this clear and potentially include experiments for other possible causes of forgetting.  We would appreciate it if the reviewer could give advice on how such an experiment can be set up. Regarding the 2nd example, client-local training essentially is a single training process of a deep learning model, therefore, any possible forgetting in a deep learning training can happen during the client local training, which makes it not specific to the intricacy of FL.
> - In the same line, the paper defines, measures and tests local forgetting. Later on, it concludes that some amount of it is necessary for learning. This conclusion is valuable, but this nuance is not reflected in the introduction nor in the motivation of the manuscript.
>     - Thank you for this feedback, we will improve the manuscript, to reflect that.
> - There is little discussion of the public data used by the algorithm until Section 5.1. Even then, I am left with questions regarding how it can affect the forgetting behavior. What distribution does this data need to be drawn from? Can it exacerbate forgetting if drawn from the wrong distribution?
>     - *Response:* Addressed in shared response
> - I am surprised that several baselines did not converge for FEMNIST in Table 1. This is a fairly simple benchmark that should achieve good performance with a CNN.
>     - *Response*: We found that in general FL experiments are tricky and very sensitive to the experiment setup, such as client selection, and number of clients. Can you please refer us to similar experiments with the same setup, where FL algorithms converged to better results?
>
> ## Questions
>
> - I found Figures 2 and 5 confusing. I'm not sure what the colors refer to, and why Flashback is performing better according to these figures. Please clarify.
>     - Response: The green color is to distinguish Flashback from the other baselines. The intensity of the color in the heatmap reflects the accuracy. Flashback is performing better in Figure 2 because it shows less drop in accuracy (forgetting) from the first row (global model at round t-1) to the rows in the middle, which represents the accuracy of the clients that participated after their local training. Furthermore, from the rows in the middle to the last row (global model at round t) Flashback shows better aggregation results, with less forgetting than the other baselines.
>     As for Figure 5, we show that Flashback is more consistent across rounds because it has fewer drops of accuracies of the global model per-class accuracy over all the rounds, which indicates forgetting. Thanks for the feedback, we will revise the figure and update the figures for more clarity in the revised manuscript
> - In line with the connections between distillation and algorithmic churn, and with other possible causes of forgetting, future versions of the manuscript would benefit from studying forgetting at the example level (see [2]) for a given test dataset at the server.
>     - Response: That is a good suggestion thanks. But can the reviewer please clarify the question here?

---

### Author Response · Authors · 2023-11-17
**Public dataset**

Again we would like to sincerely thank all the reviewers for their time and effort. We appreciate all the feedback on the paper, we we will use it to improve our work.

**This comment addresses common feedback from the reviewers about the public dataset.**

# Public dataset

There are many works that suggest and propose using a public dataset, for example [1-9]. Flashback assumes the availability of a very small public dataset, that is class (label) balanced. The practicality of a public dataset is under-studied, however, it is fair to assume that an FL system owner (Google, Meta, Apple, and other big tech companies), has the ability to collect a small dataset that is labeled. Therefore, we think our assumption of the availability of a very small public dataset is reasonable. Moreover, in many works the public dataset is set to be an entire dataset such as using CIFAR100 as the public dataset, while in our case, we sampled 2.5% of the training set to create a public dataset (1250 data points).

Moreover, as suggested by many of the reviewers, we will conduct experiments using imbalanced public datasets, and public datasets with various sizes that will be added to the revised manuscript.

[1] Lin, Tao, et al. "Ensemble distillation for robust model fusion in federated learning." *Advances in Neural Information Processing Systems* 33 (2020): 2351-2363.

[2] Bistritz, Ilai, Ariana Mann, and Nicholas Bambos. "Distributed distillation for on-device learning." *Advances in Neural Information Processing Systems* 33 (2020): 22593-22604.

[3] Yang, Qiang, et al. "FedMMD: Heterogenous Federated Learning based on Multi-teacher and Multi-feature Distillation." *2022 7th International Conference on Computer and Communication Systems (ICCCS)*. IEEE, 2022.

[4] He, Yuting, et al. "Learning critically: Selective self-distillation in federated learning on non-iid data." *IEEE Transactions on Big Data* (2022).

[5] Cheng, Sijie, et al. "Fedgems: Federated learning of larger server models via selective knowledge fusion." *arXiv preprint arXiv:2110.11027* (2021).

[6] Hu, Li, et al. "MHAT: An efficient model-heterogenous aggregation training scheme for federated learning." *Information Sciences* 560 (2021): 493-503.

[7] Lee, Sangho, KiYoon Yoo, and Nojun Kwak. "Asynchronous edge learning using cloned knowledge distillation." (2020).

[8] Li, Yiying, et al. "FedH2L: Federated learning with model and statistical heterogeneity." *arXiv preprint arXiv:2101.11296* (2021).

[9] Li, Daliang, and Junpu Wang. "Fedmd: Heterogenous federated learning via model distillation." *arXiv preprint arXiv:1910.03581* (2019).

---

### Meta-Review · Area_Chair_Y7ep · 2023-12-10

**Metareview:**

The paper presents Flashback, a novel Federated Learning (FL) algorithm to address the issue of forgetting in FL due to data heterogeneity. It introduces a new metric for measuring forgetting and integrates dynamic distillation in both local and global update phases.

Strengths:

- Addresses a significant issue in FL.
- Empirical results demonstrate the superiority of Flashback over baselines.

Weaknesses:

- Insufficient discussion on the role and selection of public data used, which potentially requires more thorough reviewing after providing.
- The assumption that the distribution of public data is the same as training data, which is a significant limitation.
- Lack of a systematic exploration of various factors affecting forgetting in FL.

Decision:
The paper shows potential with its novel approach and significant contributions but also has notable weaknesses. Overall, I agree with the reviewers that the paper is not ready for publication.

**Justification For Why Not Higher Score:**

The weaknesses listed were not resolved during the discussion indicating that a more thorough review of the revised manuscript would be needed.

**Justification For Why Not Lower Score:**

N/A

---

### Decision · Program_Chairs · 2024-01-16

Reject